# Research Scenarios of Autonomous Vehicles, the Sensors and Measurement Systems Used in Experiments

**DOI:** 10.3390/s22176586

**Published:** 2022-08-31

**Authors:** Leon Prochowski, Patryk Szwajkowski, Mateusz Ziubiński

**Affiliations:** 1Institute of Vehicles and Transportation, Military University of Technology (WAT), ul. gen. Sylwestra Kaliskiego 2, 00-908 Warsaw, Poland; 2Łukasiewicz Research Network—Automotive Industry Institute (Łukasiewicz-PIMOT), ul. Jagiellońska 55, 03-301 Warsaw, Poland; 3Doctoral School, Military University of Technology (WAT), ul. gen. Sylwestra Kaliskiego 2, 00-908 Warsaw, Poland

**Keywords:** research and test scenarios, specialized test tracks, proving grounds, automated and autonomous vehicles, experimental research, ADAS, perception sensors, lidars, cameras, radars

## Abstract

Automated and autonomous vehicles are in an intensive development phase. It is a phase that requires a lot of modelling and experimental research. Experimental research into these vehicles is in its initial state. There is a lack of findings and standardized recommendations for the organization and creation of research scenarios. There are also many difficulties in creating research scenarios. The main difficulties are the large number of systems for simultaneous checking. Additionally, the vehicles have a very complicated structure. A review of current publications allowed for systematization of the research scenarios of vehicles and their components as well as the measurement systems used. These include perception systems, automated response to threats, and critical situations in the area of road safety. The scenarios analyzed ensure that the planned research tasks can be carried out, including the investigation of systems that enable autonomous driving. This study uses passenger cars equipped with highly sophisticated sensor systems and localization devices. Perception systems are the necessary equipment during the conducted study. They provide recognition of the environment, mainly through vision sensors (cameras) and lidars. The research tasks include autonomous driving along a detected road lane on a curvilinear track. The effective maintenance of the vehicle in this lane is assessed. The location used in the study is a set of specialized research tracks on which stationary or moving obstacles are often placed.

## 1. Introduction

Currently, there are a large number of publications on autonomous vehicles. Most of these papers, along with their included research and discussions, deal with technological aspects. The dominant topics are vehicle control, avoiding road obstacles, and sending information about the current situation of the vehicle and possible threats. This is happening because these vehicles are in a phase of intensive development. A very important stage of development work is model and experimental research.

The test vehicles, often conventionally referred to as autonomous in many publications, perform strictly planned tasks during research. Thus, they operate as automated vehicles. In this paper the abbreviation AV will be used, which should be considered as an automated or autonomous vehicle. According to the SAE J3016 standard [1,2], autonomous vehicles (AV) are accepted cars classified between levels 3 and 5.

However, experimental research related to AVs is in an initial state. This is confirmed by a small number of studies of complete control systems for such cars. For example, there are studies that analyze environment perception systems and the vehicle control is investigated separately. Research is still at the stage of gathering knowledge about the performance of different systems under different conditions. Research plans are visible and adapted to the current cognitive needs of the research organizer. This situation is the result of a lack of recommendations and standardized guidelines for the scenarios used in such studies. There is a big problem and binding arrangements (e.g., EU, USA) for the organization of safe AV testing in road traffic is sorely needed [3]. Particularly, the requirements for the scenarios used in such studies have not been specified. Difficulties regarding these scenarios also result from the fact that, in the planned research, local and readily available technical resources and available equipment are first taken into account. In autonomous vehicles, an infinite number of variants related to vehicle behavior arise as a result of autonomous decision-making by the control system. Hence, there may be a very large number of research scenarios. Particular difficulties arise in the organization of research in critical traffic situations [4,5].

The main difficulties that are faced during the creation of research scenarios with a planned course (that is, in automatic vehicle control mode) can be summarized as follows:-the number of systems to be checked at the same time is significant and the vehicles have a very complex structure;-the number of places where conducting research is possible is limited;-the trajectory of the tested vehicle may deviate significantly from the planned trajectory;-the legal regulation of AVs’ participation in traffic is inconsistent and unclear in different countries.

A complication in the organization of the tests also arises from the fact that, in traffic, all the vehicle’s systems will be operating simultaneously or in a sequence that is difficult to plan [6].

The development of scenarios for experimental research related to road traffic can be based on the results of previous studies (available in significant numbers), namely:-the results of model tests and computer simulations in which the behavior of the vehicle in virtual space can be safely observed (technologies: MIL [7], SIL [8]);-the results of analysis focusing on cooperation between computer models and the systems of a real vehicle (technologies: HIL [8]);-research results of complete vehicle actuators in combination with road tests and the simulation environment (technologies: VIL [7,9]);-results of the validation of the human–machine interface through the interaction between the driver and the autonomous vehicle (technologies: DIL [7]);-research results of the vehicle at the proving grounds (closed area, dedicated track).

There is a general consensus that the scenarios of such research should result from the observation of road traffic and drivers’ behavior [10,11,12].

Currently, there are significantly more simulation studies than experimental studies. In [13,14], the benefits of implementing mixed/hybrid research are described. In such a method, signals for perception systems during experimental research are supplemented by signals generated from computer simulations. This allows for better repeatability of test conditions and realization of more complex scenarios. On the other hand, in [15], the concept of so-called parallel research was presented, in which a real autonomous car reacts to signals from movies shown on screens placed around it. This ensures the research has high repeatability due to the possibility of reproducing the same scenarios.

The above-described research and all other experimental research related to AVs has been conducted according to specific scenarios. The experimental research scenario requires a lot of information to be established:-aim of the research;-test vehicle (platforms) data and the method of their preparation and configuration;-the use of the environment (surroundings, test track) in which the research will be conducted;-a set of elements and additional objects (complementary) which are required to be included in the research;-initial conditions and the planned course of the study (test sequence);-measuring equipment, its tasks, and method of use;-the quantities necessary to measure, the level of accuracy of the measurements, and the method of evaluation of the results;-safety procedures.

Similar findings are included in scenarios in current normative research regulations, e.g., SAE [16], Euro NCAP [17], and EU [18].

The aim of the article is to present the main information about AV research scenarios, and the study tasks undertaken in this area, in an orderly manner. The collected material aims to systematize the information that will facilitate the planning, preparation, and execution of experimental AV research. The developed material will allow test preparation time to be shortened, the repetition of already performed research to be avoided, and the number of wrong decisions made at this stage to be reduced (including selection of the research aim, measuring and recording equipment, localization devices, etc.). The systematization of available information on normative and non-normative scenarios will facilitate their analysis and the assessment of their applicability when planning AV research.

The material gathered in the article will help identify the following:-research tasks that currently dominate;-elements to be included in the AV research scenario;-which normative documents are useful for organizing research;-what measuring and recording equipment is necessary.

This paper presents the currently used normative scenarios of experimental AV research. Scenarios used by various research centers will be characterized, in which the plan and course is adapted to the current needs of researchers (nonnormative scenarios). Attention will also be paid to the technological aspects of the research scenarios, as well as the measurement systems and sensors used in the research.

On the basis of the literature analysis, a hierarchical scheme of information necessary for the creation of AV research scenarios has been built. The layout of this scheme is shown in Figure 1. The article presents individual elements of this set.

There is a lack of publications focused on the preparation and organization of AV research. In publications containing the results of experimental studies, the authors mainly focus on the results of these studies, and information about their preparation and scenario is limited. This additionally justifies the need to gather and systematize knowledge in this area. During the preparation of the article, our own database of scientific publications about AV research was used. The database was created over many years of research carried out at the Research Network Łukasiewicz—Automotive Industry Institute in Warsaw on the improvement of AVs.

The collection of research material for the article was based on the usage of scientific publications databases such as Springer, Science Direct, MDPI, Taylor & Francis, IEEE Xplore, ResearchGate, and SAE International. Moreover, online resources from many organizations were analyzed separately, including Euro NCAP, NHTSA, ISO, EU, UNECE, and IIHS. The result of tagging topics of interest and research areas was also a set of keywords. This was the starting point for the search of publications in the databases. Selected articles were categorized in three areas (AV research scenarios, technical aspects of the AV research, and sensors and measuring equipment). These areas are the main content of the article.

## 2. Objectives and Scope of AV Research

Experimental AV research considers many multidisciplinary issues related to the road traffic control process. One of the most frequently studied issues is the problem of object detection by the environment perception system. An example of such research conducted by the Łukasiewicz Research Network—Automotive Industry Institute in Warsaw is shown in Figure 2. Researchers analyze the time it takes for AVs to detect a suddenly appearing obstacle (red car). The obstacle comes from behind the car leaving the road intersection (blue car). Research was carried out for various driving velocities, the shape and color of the obstacle, as well as for varying illumination (day, night). Many authors present publicly available and proprietary models for object detection. Obstacle classes such as cars [19], pedestrians [20,21], bicyclist/motorcyclists [20], traffic signs [22], and others [23,24] are recognized. These models are based on vision sensors in the form of RGB cameras, from which fragments of the surrounding image are analyzed by neural network models. Characteristic features of a given object are sought. Perception systems based on the use of several devices are increasingly being considered, which makes it possible to eliminate the disadvantages of some sensors. The most common way to combine information is sensory fusion of the lidar and camera signals [23,24,25]. The detected objects make it possible to generate the output signal from the perception system, which is the basis for further decision-making processes in the AV control system.

A large amount of the papers published in this area discuss the problem of road edge and lane detection by AVs [26,27,28]. Regarding this issue, the image from the camera of the perception system is most commonly used. Usually, the edges of the road, lines limiting the lane of the road, or free spaces in front of the vehicle are usually marked. Figure 3 presents a part of the research conducted by the authors. The result of the lane and road edge detection by the AV’s perception system is presented while it drives on a three-lane road. The road edges determined by the perception system are marked in red and yellow (left and right edge, respectively). The colors green and blue indicate the edges of the occupied lane (left and right edge, respectively). Despite the fact that parts of the lines are obscured by other vehicles, the perception system estimates their location in the missing areas.

Another research problem is avoiding obstacles on the road. Planning an avoidance maneuver requires information from the perception system about the position and dimensions of the obstacle. There is a separate need to define the area in which the AV can continue driving after passing the obstacle (detected road lane, free spaces, road edges). Based on this, the decision-making system generates a trajectory for further driving (path planning). Usually, these are trajectories safely avoid obstacles in road traffic [29].

Maintaining the planned path strongly depends on the curvature of the path and driving velocity. These factors influence the lateral force values (centrifugal force), which causes sideslip and lateral skidding of the tires. The result is a deviation of the real trajectory from the planned path. This deviation can be measured as shown in Figure 4. It is calculated as:(1)$$\Delta y_{MA}=y_{M}\left(x\right)-y_{CA}\left(x\right);\Delta \psi_{MA}=\psi_{YM}-\psi_{CA},$$
where the individual parameters are the geometric quantities indicated in Figure 4.

This is exemplified by the planned path $y_{M}\left(x\right)$ and car driving trajectory $y_{CA}\left(x\right)\;$ in Figure 5a. The planned path is the result of the control system’s calculation based on information from the perception system. Driving trajectory is the actual trace of the vehicle’s movement (the car’s response to the control process) on the distance of avoiding the obstacle. This distance in the drawing is 30 m long. Figure 5b,c show examples of deviations of driving trajectory from the planned path for several driving velocities as part of the suddenly appearing obstacle avoidance maneuver. These are the results of simulations that the authors partially showed in [4]. There are visible differences between the planned path and the actual AV trajectory. These differences increase with increasing driving velocity.

The figures show the influence of driving velocities on driving trajectory deviations from the planned path. The lateral forces (centrifugal forces) increase with increases in driving velocity on the curvilinear path, which causes sideslip and lateral skidding of the tires. As a result, an increase in the lateral deviation $\Delta y_{MA}$ of the driving trajectory from the planned path is observed and increased angular deviation of the longitudinal vehicle axis $\Delta \psi_{MA}$. This issue is one of the most important problems in experimental AV research.

AV’s contribution to road traffic is based on many activities. These include:-recognition of the road situation;-planning a collision-free path;-current assessment of the real trajectory in relation to the planned path and making the necessary corrections.

Path planning is based on:-navigation and digital maps of the driving area;-location of the vehicle in relation to the road infrastructure;-environment perception to recognize the situation around the vehicle.

The problem of planning and maintaining a planned path is often the aim of experimental research or is part of other research in which the planned path is an indirect factor.

The primary task during AV movement is performed by the perception system. The result should be at least the detection of the road lane edge or its centerline and an estimate of the relative position of the car in that lane. This is the dominant problem of experimental research in the aspect described here.

## 3. Experimental Research Scenarios

### 3.1. Scope and Features of Typical AV Research

Experimental AV research scenarios found in the literature have a certain set of features. These often depend on the aim of the research. The main features of the AV research scenarios, in terms of self-driving and the effectiveness of lane-keeping by the vehicle, are summarized in Table 1. These are automated driving scenarios, i.e., following a planned path or a preset road lane.

Table 2 shows examples of research scenarios that aim to analyze the behavior of a vehicle in urban areas, namely:-effectiveness of road line (lane boundary) detection;-the planning of a safe path for avoiding a slower moving vehicle.

The next scenario concerns research into avoiding suddenly appearing obstacles (Table 3). Thus, this research is different to the previously described scenarios.

### 3.2. Normative and Unified AV Test Procedures

Normative AV test scenarios and procedures are used to test finished products and automation systems, e.g., for homologation. An example of a type of system for which normative testing is carried out is ADAS. Driver assistance systems include FCW, AEB, LKA, or ESP. ADAS are maximum level 2 solutions according to SAE [2,35,36]. They form the basis of higher-level systems. Such test scenarios have a number of features that should also be found in the research scenarios considered in this paper. Therefore, the results of a review of normative AV testing scenarios and procedures are presented. These are defined by organizations:−EU [18];−UNECE [37,38,39];−ISO [40,41,42];−Euro NCAP [17,22,43];−NHTSA [44,45,46];−SAE [16,47];−IIHS [48,49].

Normative scenarios allow for repeated testing of entire vehicles and their systems. Unified procedures, equipment, and methods of evaluation of test results make it possible to compare the various solutions used in AVs. This makes it easier to decide on appropriate directions of development for active safety systems in vehicles. A structured overview of normative testing scenarios is presented in Table 4.

Normative test scenarios are focused on the testing of finished systems in AVs. These are most often certification, homologation, and ranking tests (these tests are not of a cognitive nature). The analysis of normative test scenarios shows that the largest part of studies is currently carried out for driver warning, emergency braking, and lane keeping systems. The effectiveness of these systems is assessed in terms of vehicle response to the detected threat (only the effect of the system’s reaction is assessed). This may include the threat signals for the driver or the effectiveness of an automatic defensive maneuver. Normative scenarios are mostly carried out on test tracks/proving grounds with the necessary infrastructure. Selected scenarios enable tests to be conducted on public roads. In these cases, critical situations are ignored and AV control is not implemented. The analyzed normative scenarios indicate that AV tests are currently focused on critical situations in road traffic. This is the basis for building nonnormative research scenarios. These nonnormative scenarios should focus on the cognitive aspects of autonomous vehicle operation. The reasons for the AV’s reaction to the threat should also be assessed (e.g., inference about the behavior of the perception or control system).

### 3.3. Nonnormative Research Scenarios

Nonnormative test scenarios are created to conduct experimental research related to the development and analysis of the performance of AV systems. Therefore, these are scientific pieces of research of a different basis than the normative test scenarios described in the previous section. These scenarios are focused on the cognitive aspect of experimental research. Such scenarios are also needed during component/system research, e.g., object detection by the perception system [67,68,69], path planning systems [19,33], and road line/edge detection [26,27,70]. Nonnormative AV test scenarios were analyzed. These scenarios were organized and grouped based on the scope and aim of the conducted experimental research. These scenarios take into account:−AV path planning [33,34], where the goal is usually to develop a system for determining a non-collision path or smoothly avoiding slower/stationary objects;−following the previous vehicle, e.g., as part of the movement within two circles [19];−road line detection and traffic lane detection [26,27,70], where the aim is to detect an area where the driving process can be continued by the AV;−AV lane keeping:○driving with high velocities on highways [30,32];○driving on a racing track on a wet and dry surface and different strategies of AV path planning [31].−emergency braking before an obstacle [34,71,72], where the aim is to minimize the risk of collision;−avoiding a suddenly appearing obstacle in a critical situation where it is not possible to stop the AV [34,72];−2D object detection, localization, and tracking:○car detection [70,73,74] and distance estimation to other vehicles [27,75];○pedestrian detection [20,21,69];○bicyclist/motorcyclist detection [20,73,74].−3D object detection on the road [72,76] using lidar and sensor fusion (e.g., lidar and camera), where the aim is to obtain information about the obstacle in the spatial coordinate system:○car detection [19,75,77];○pedestrian detection [23,24,78];○bicyclist/motorcyclist detection [23,25,73];○cone detection [67];○detection of walls, trees, bushes, and poles [78];○detection and tracking of unknown objects [79].

The analyzed nonnormative research scenarios indicate that currently single AV systems (components) are the main directions of development within experimental research. Few publications discuss scenarios where a complete AV system is analyzed in terms of its control. Some focused on critical situations and these are conducted on tracks. Algorithms such as lane keeping, braking, or obstacle avoidance are being investigated. The number of these publications is growing, but it is still small in relation to the research of single AV systems (components). This can be influenced by the cost of conducting research into complete AV systems. They are also characterized by the complexity of preparation for the research scenario and a very large number of variants. The issue of perception systems focused on object detection and line/edge detection on the road is widely researched. A significant number of research scenarios for these systems are evident. They are often carried out in road traffic because they do not reduce the level of safety for other road users. These studies are focused on calm driving without the risk of critical situations.

## 4. Technical Aspects of AV Research

The organization of research according to specific scenarios requires meeting the technical requirements contained therein. Technical aspects in the research scenarios include the characteristics of the research environment and the characteristics and parameters of the vehicle. The relevant issues in this area will be presented below.

### 4.1. Research Vehicles

Research according to the described scenarios is usually carried out with the use of passenger cars [31]. The movement of the vehicle during the research is automated or controlled by the driver. Depending on the subject and scope of the research, the following are also used:−buses [79];−trucks [67];−microcars (single person) [34];−go-karts [32];−research platforms (autonomous systems carriers) or physical models of the car at a reduced scale [80,81].

The use of reduced-scale car models allows for significant reductions in research costs, the space required for their realization, as well as the risk of possible damages in case of a defective realization of the scenario. It should be noted, however, that conducting research on physical models may not provide complete information on the dynamics of real car motion, and the problem of scaling vehicles is a complex and problematic issue [82,83,84].

### 4.2. Research Site

Many countries still do not allow the testing of autonomous vehicles on public roads. Therefore, in addition to limited road fragments, special places are prepared for the realization of experimental research of autonomous cars, such as:−the proving grounds for testing and validation of AVs in Zalaegerszeg (Hungary) and Immendingen (Germany) [85,86];−proving grounds Mcity (Almono (Uber ATG)) and Kcity [13,87,88];−proving grounds containing an oval track (such as KATRI—Korea Apparel Testing and Research Institute) [30];−tracks used in car racing (Monza in Italy) [31];−fragments of road infrastructure excluded from public traffic which have been specially selected for conducting research according to specific scenarios [89];−polygons using an unstructured environment [90,91], which sometimes lacks road infrastructure elements such as vertical and horizontal signs.

The number of public roads where research is allowed is small, but has been increasing recently [92].

### 4.3. Infrastructure Elements and Other Objects

The realization of experimental research often requires the use of additional objects. Their presence within the range of the car’s perception system is the basis for the operation of the control system. In road traffic, these include other vehicles, pedestrians, cyclists, or fixed obstacles. If the course of the research is not related to the realization of critical maneuvers, the scenarios depend on the state of road traffic. Research results according to such scenarios are available in [25,67,70].

Due to limitations surrounding the possibility of conducting research on public roads, it is necessary to conduct research in a closed area. In such an area, real objects (e.g., other road users) are replaced with various types of dummies, decorations, and physical models. Table 5 summarizes examples of artificial objects used in automated vehicles test scenarios, e.g., in the role of obstacles in their planned path.

Figure 6 shows the objects (cf. Table 3) used in AV experimental research. In [34] boxes and cardboard boxes were used to mark the road lane, whose arrangement around the vehicle was the basis for controlling AV movement (Figure 6a). Figure 6b shows a soft passenger car target after mounting on a moving platform. The target was used in scenarios of avoiding a suddenly appearing obstacle [17,28]. For example, the car model moving on a collision path with AV, forced a change of AV driving path and trajectory.

Dummies of pedestrians, children and bicyclists are used in a similar role (Figure 6c [43] and Figure 6d [56]). Figure 5d also shows an example of an obstacle in the AV path, built in the form of a soft wall covered with a material with reflective properties to facilitate detection by the vehicle perception system [56].

Conducting experimental AV research requires the provision of appropriate facilities and technical conditions. The AV system is located on the research vehicle, which is usually a passenger car. However, there are test scenarios where other vehicles, go-karts, or reduced-scale vehicles are used. Due to the limited possibility of conducting research in road traffic, most AV research is carried out on test tracks and proving grounds. They are usually equipped with road infrastructure to replicate actual road conditions. It is necessary to use dummies, models, and mannequins of other road users, including pedestrians, children, bicyclists, or other vehicles. Their role in research varies, but usually their presence in the environment is the basis for AV control.

## 5. Measuring Systems, Sensors, and Measurement Ranges

The considered AV research scenarios describe the methods of testing individual systems of the autonomy system, e.g., control during uninterrupted movement (lane keeping on highways) and during critical situations (active safety systems). This type of experimental research requires many sensors. These sensors can be divided into sensors for the AV’s perception and control systems, as well as devices recording the results of research (measuring equipment).

### 5.1. Sensors Used for Perception and Control Systems

AV perception systems are an integral part of any control system, regardless of their level of sophistication. Single sensors are used in simple systems [27,44,55]. Many different sensors are used for more advanced solutions, which receive signals that complement each other and provide complementary information [24,25,79]. Sensors in AV systems are devices that generate and transmit information about the traffic environment for further analysis (e.g., information about road infrastructure and locations of other objects/obstacles). They are also sensors for measuring the vehicle motion parameters necessary for driving planning and control (the AV’s position, vehicle velocity, acceleration, angular velocities, etc.).

The sensors of the perception systems are mainly vision sensors (cameras) recording RGB images [31,32,70], radars using radio waves [55,57,71], and lidars based on laser beams [24,25,33]. Most solutions based on vision sensors use single-lens cameras that provide a 2D image for the road situation analysis system [24,32,72]. There are also vehicles and systems that use stereo cameras [31,67,74]. These sensors provide an image from two lenses, and this creates an image with depth. This solution enables 3D observation of the surroundings. Both types of cameras are relatively low-cost, which allows them to be widely used [20]. More complex solutions use thermal imaging cameras to observe the AV’s surroundings [20,95]. These sensors provide information about the temperature of the surroundings. This makes it possible to detect objects and obstacles based on the temperature gradient. The disadvantage of these sensors is their high price, low resolution, and slower image refresh rate compared to RGB cameras [20]. A summary of the advantages and disadvantages of sensors in AV perception systems is presented in Table 6.

Radars are commonly used sensors which provide information about obstacles and vehicle movement parameters. Radars are used for emergency braking systems [55], active cruise controls, and for object detection in a vehicle’s blind spots [44]. The advantage of this type of sensor is their operation at high driving velocities, especially over long ranges. The maximum range of the radar can be over 150 m [72] (at a maximum of 25 m for the Stereolabs ZED camera [31,96]). The disadvantage of radars compared to cameras is the small amount of information acquired about the spatial dimensions of objects. Radars generate a scan of the environment in the form of a point cloud based on reflection of a radio wave. Such a radar scan contains a small number of points, which makes it difficult to recognize objects [20]. This disadvantage is eliminated by lidar sensors (laser sensors), which are characterized by a large number of layers that scan the area around the vehicle. Lidar sensors generate a scan of the environment around the vehicle in the form of a point cloud. This enables the object and infrastructure elements to be detected, along with their classification. The collected information is similar to the depth image of a stereo camera, but with a much greater scanning range. They are characterized by high accuracy in terms of object localization (e.g., 3 cm—typical measurement accuracy for Velodyne VLP-16 lidar). The disadvantage of lidars is the very high price (especially for sensors with a large number of scanning layers), which in many solutions prevents their widespread use [20].

AVs are equipped with additional sensors for control systems, namely:−GPS [19,33,74];−GNSS RTK [31,75,95];−IMU [19,34,95].

The GPS/GNSS system enables localization of the vehicle’s position using a satellite signal. This is a very common sensor. It is usually used for global path planning of the AV. The disadvantage is the low accuracy of the localization—usually 1–10 m. An extension of this system is to complement GPS/GNSS with RTK. It is a GPS system that allows precise determination of the position of the vehicle in real time. In addition to the GPS in the car, this kit uses an additional signal from a stationary reference station, which also receives a GPS signal. The reference station sends the data about position correction to the receiver in the car. This increases localization accuracy to approximately 4 cm.

The IMU sensor is a device that integrates an accelerometer, gyroscope, and magnetometer. This sensor enables the determination of vehicle acceleration, angular position, and angular velocity. IMU is very often found as a supplement to GPS and GNSS RTK systems.

### 5.2. Sensors in Research Scenarios

#### 5.2.1. Cameras

In experimental research, AV cameras are used to detect lines on the road [27] or during lane keeping [31,32,79]. Their widest area of application is object detection on roads. There are research scenarios that take into account 2D detection of objects and obstacles [20,70,75]. There are separate scenarios for the detection and localization of objects in 3D space, based on the sensory fusion of several perception sensors (cameras with lidars or radars) [24,67,77]. Cameras are used in emergency braking systems and collision warning systems about an obstacle (e.g., in front of another vehicle, pedestrian, or bicyclist) [51,56,72]. They are also used for issues related to monitoring and warning about objects in the blind spot of the vehicle [44]. Cameras from different manufacturers are used in AV research. Camera properties are usually characterized by a set of parameters: H, V, R, f, and range.

Table 7 presents examples of the use of cameras in AV research. The research scope of the selected research scenarios has been summarized and the sensor models with selected parameters have been specified.

#### 5.2.2. Radars

The signal from the radar can be used to calculate the object’s motion parameters (e.g., position, velocity) [57,71,72]. Radars are also used in the sensory fusion of several sensors:−radar–camera–GPS fusion [72];−radar–camera fusion [51].

Long-range radars are mainly used for object detection at high driving velocities, with short-range radars used for precise maneuvers at low velocities. Radars are used in braking and obstacle avoidance scenarios [71], 3D object detection [72], and research into emergency braking systems and collision warning systems about obstacles in front of AVs (car obstacles [57], pedestrians [50], bicyclists [56]). Various manufacturers’ radars are used in AV research. Their properties are usually characterized by a set of parameters: H, f, and range.

Table 8 presents information about the radars used in AV research according to the analyzed scenarios. The research scope of selected research scenarios has been summarized and the sensor models with selected parameters have been specified.

#### 5.2.3. Lidars

Lidars create an image of the environment in the form of a point cloud [25,44]. Based on the arrangement of these points, a map of the surroundings can be built [33], road edges can be detected [34], or other objects on the road can be identified [76]. Lidars are the most commonly used sensor for sensory fusion with a camera [17,50,72]. The properties of lidars can be characterized by parameters: H, V, R, f, range, and layers.

Lidar sensors with a wide horizontal field of view (e.g., H: 360 deg) and a large number of scanning layers (e.g., Layers: 64) have high scanning resolution with a wide vertical measurement range. Reducing the number of scanning layers while maintaining the vertical measurement range limits the resolution of the sensor (reducing information about the objects). Maintaining high resolution with a lower number of scanning layers can adversely affect the observation of tall objects. Lidars with a narrower horizontal scope of observation are also used in AV research (e.g., H: 90 deg). Additionally, 360 deg sensors allow the entire environment surrounding the AV to be scanned with a single device (with a sensor that is usually mounted on the roof of the vehicle). The disadvantage of this solution is the lack of information about obstacles in close proximity to the AV due to insufficient vertical measuring range. The use of sensors with a smaller horizontal measuring range requires the installation of several devices around the vehicle to observe the entire environment. This increases the vertical measuring range in areas close to the vehicle. The disadvantage is the high cost of many devices. Increasing the number of scanning layers, resolution, and horizontal measurement range of the sensor/sensors increases the size of the collected files. This may have an adverse effect on the time taken by the AV’s perception system to detect objects.

Table 9 summarizes the lidars used in experimental AV research for different scenarios. The research scope of selected research scenarios has been summarized and the sensor models with selected parameters have been specified.

#### 5.2.4. Other Sensors

Table 10 summarizes the types and models of sensors other than those described in Table 8 Table 9 andTable 10. These are sensors and devices used in the AV localization and control process. In [19,33], the GPS sensor is used in path planning and object detection scenarios. A similar application has the GNSS RTK Swift Navigation sensor in [31,75]. This sensor is used in lane keeping, localization, and object tracking scenarios. In [74], a NovAtel SPAN-CPT GPS/INS device was used in object detection and tracking scenarios (e.g., pedestrians, cars, bicyclists). The GPS/INS sensor is used in sensory fusion with the lidar to improve the accuracy of the AV’s position in space relative to the 3D detected obstacle. The IMU sensor for measuring acceleration and angular velocity was used in [33,34]. There, scenarios related to path planning, braking, and obstacle avoidance were considered. This type of sensor is useful in scenarios where the influence of AV dynamics on the process of its control is analyzed.

The analysis of the sensors used in the perception and control systems shows that the most commonly used solutions in AVs are based on cameras (2D detection) and lidar sensors (3D detection). Many scenarios use sensory fusion of these two sensors. This enables the elimination of the disadvantages of individual sensors, as well as the extension of the measurement capabilities of the AV’s perception system. Lidars that fully scan the environment around the vehicle are increasingly used. This enables the use of a single sensor on the vehicle. The use of sensors with an increasing number of scanning layers is visible in AV research. In experimental research, radars are mainly used in normative scenarios as the basis for ADAS solutions (e.g., AEB). Additionally, in ADAS and in nonnormative scenarios, cameras are used to test LKA, LSS, or perception systems which detect lines or road signs. There is a growing number of research results which take into account sensors such as GPS and GNSS RTK, enabling precise determination of the AV’s position (e.g., in relation to an obstacle). There is increasing research into the impact of AV dynamics and kinematics in terms of its control. For this purpose, IMU sensors are increasingly being used to measure quantities such as acceleration or angular velocity of the vehicle.

### 5.3. Measuring Equipment

Experimental AV research uses a wide range of sensors which are necessary for perception and control systems. Verification of the operation of the AV system requires external measuring equipment, which will enable the parallel measurement of selected physical quantities (e.g., position, velocity, and acceleration) and other signals. These recorded data are the basis for assessing the performance of AV systems in the considered research scenario.

The measuring equipment is only used to record the course of the research and does not affect the decision-making or AV control process. The measuring equipment used depends on the considered research scenario; however, several characteristic areas of measurement can be identified:−waveforms related to the movement of the vehicle/obstacle (e.g., position and acceleration);−information and warning signals from the AV system interface;−course of the experiment (image from internal and external cameras installed on vehicle);−signals related to the vehicle control process.

The measurement of waveforms related to the movement of the vehicle or obstacle for a given scenario is related to the measurement quantities such as the position of objects, driving velocity, acceleration (linear, angular), angular velocity, etc. For this purpose, information transmitted by GPS/GNSS, GNSS RTK, and IMU sensors is used [95]. In scenarios related to research into the emergency braking system and collision warning system, the ADMA or DGNSS device has been used to measure waveforms [50,51]. In a similar scenario in [55,56], waveforms from the wheel’s angular velocity sensor and the GNSS RTK module were also measured.

Normative scenarios specify the range of vehicle dynamics parameters and the obstacle parameters necessary for registration during the tests. Either the sampling frequency (e.g., min. 100 Hz) or the accuracy of the measurements is indicated (velocity 0.1 km/h, position localization 0.03 m, angular position 0.1 deg, angular velocity 0.1 deg/s, acceleration 0.1 m/s^2^, steering wheel angular velocity 1.0 deg/s). These requirements are developed for scenarios related to emergency braking systems, collision warning systems [17,43], and testing of lane keeping systems [28].

In [44] (a research scenario for blind spot monitoring systems) and [75] (a 3D object detection scenario), for the measurement of the AV’s movement and obstacle parameters, OXTS equipment was used. Specifically, the RT3000, RT Hunter, RT Target, and RT Range were the devices used.

The experimental research of systems, the main task of which is to warn the driver about dangers, requires the registration of messages appearing during a critical situation (e.g., FCW or LDW systems). The normative test scenarios for these systems are described in [17,28,43]. Visual and sound-based warning signals for the driver are recorded indirectly using cameras located inside the vehicle. In [50,51] (an emergency braking/collision warning system research scenario), GoPro cameras were used to record visual warning signals on the AV’s dashboard and warning sound signals. In [44], warning signals for the driver (sounds and visuals) related to an object in the blind spot were recorded. In [22], visual signals about the recognition of a road sign were recorded.

Cameras are used to record the progress of an experiment. They can record the course of the research inside the vehicle or can be mounted on the AV’s body and record the surroundings of the vehicle. For this, SLR cameras can be used, which can additionally be activated at a defined point in time in order to synchronize data recording [50,51] (emergency braking/collision warning system research scenario).

The last area related to measuring the course of research is signals generated by the AV system for controlling vehicle movement. In [55,56], signals such as the value of steering wheel torque or the force on the brake pedal were recorded. This information makes it easier to evaluate the performance of the systems, especially in the case of normative testing where there is no access to the inside of the AV system. In [30], for the AV lane keeping test scenario, the values of the steering wheel’s turning angle and the steering torque were measured. The sensors used were components of the vehicles systems. The signals were read from the CAN bus of the vehicle.

In [44], the blind spot monitoring and braking scenario for the lane change assistant was considered. The braking maneuver was carried out using the Heitz Automotive Testing device. This apparatus enables the measurement of data related to braking intensity and the resulting braking deceleration.

The role of measuring equipment is crucial for quantity measurements that enable the evaluation of the course and results of experimental AV research. This is especially true of scenarios in normative testing of ADAS systems. ADAS systems do not enable measurement of physical quantities which form the basis of vehicle control. This applies, for example, to signals from the perception or decision system. Only the final result of the AV’s control system can be observed (sound or visual signal of the system interface for the driver, execution of the maneuver by the vehicle, etc.). The measurement equipment used is often characterized by high measurement accuracy, which allows the measured results to be used as a reference for signals from the AV. This mainly applies to nonnormative research scenarios.

## 6. Summary and Conclusions

Experimental research is carried out according to more and more complex scenarios. The scenario makes it easier to organize and conduct research. It contains a lot of important information, e.g., the aim and scope of research, measurement equipment, and safety procedures. The analyzed scenarios were divided into two groups, which were conventionally named:-normative and unified (e.g., SAE, Euro NCAP), which are applicable when testing vehicles and their systems in a repeatable manner (testing, inspection tests); unification of scenarios will make it possible to compare the different solutions used in AVs;-nonnormative, which were developed in various research centers; their course is adapted to the current needs of researchers. The research is of a cognitive nature and indicates the ways AVs can be improved and developed.

There are a significant number of scenarios that focus on the research of AV systems. The leading area of research regarding components is perception systems. These systems are dominated by two methods of object detection: 2D and 3D detection. Systems based on vision sensors use single-lens cameras that provide 2D images. On the other hand, systems with stereo cameras provide a spatial image from two lenses. This solution enables 3D observation, typically at distances of up to 30 m. An advantageous but costly solution is to use information from radar and lidar (especially with a large number of scanning layers). This arrangement enables the spatial localization of objects and road infrastructure at greater distances than a stereo camera. It makes it easier to detect obstacles and increases the accuracy of object detection, determining their distances to AV and dimensions. Increasingly, at the stage of processing information from sensors, sensory fusion methods (combining information from cameras and lidars) and neural network models are used for object detection (obstacles) in road traffic.

AV research has been carried out in dedicated areas of road infrastructure. In such an area, real objects (e.g., other road users) are often replaced by various types of dummies, targets, and physical models. Their location is described in test scenarios, and the results have an impact on the effectiveness of the AV control process, including obstacle avoidance. This process affects the realization of the planned path and road safety. The AV control system requires current information about the position of the vehicle and the road lane. The investigated scenarios showed that this process is based on the use of information from sensors and systems, such as GPS, GNSS RTK and IMU sensors/systems.

The current experience, resulting from the analysis of AV tests, indicates the need for further standardization of research procedures. However, the limitations in the conducted analysis result from several factors:-there is a lack of publications reporting experimental research related to AVs;-there is little information that can be obtained about the experimental research scenarios in the analyzed publications;-the lack of many legal regulations in the field of these studies means that a large part of the scenarios should be treated as individual ideas and solutions.

Expanding activities in this area will increase the possibility of comparing and summarizing experimental AV research results, which will create an increasingly complete picture of the behavior of autonomous vehicles in road traffic. The knowledge gathered in this way should significantly improve road safety and set directions for the further development of AVs. Future scenarios should also include the study of necessary changes to road infrastructure to facilitate the operation of AVs. Defining these scenarios should take into account the current capabilities and advancement level of AV systems.

This article indicates the currently dominant research tasks and methods of AV research. Another important area of systematized information is lists of standards describing standardized procedures for ADAS systems testing. In addition, an important fragment is the comparison of frequently used AV sensors with the available research topics. This makes it possible to compare the results of research centered on the use of a given sensor model to the results of other works in this field.

The designation for AV as used in this work also includes the terms contained in Regulation (EU) 2019/2144 of the European Parliament and of the Council of 27 November 2019 [98], where automated vehicle means a motor vehicle designed and constructed to move autonomously for certain periods of time without continuous driver supervision but in respect of which driver intervention is still expected or required, and fully automated vehicle means a motor vehicle that has been designed and constructed to move autonomously without any driver supervision.

## Figures and Tables

**Figure 1 sensors-22-06586-f001:**
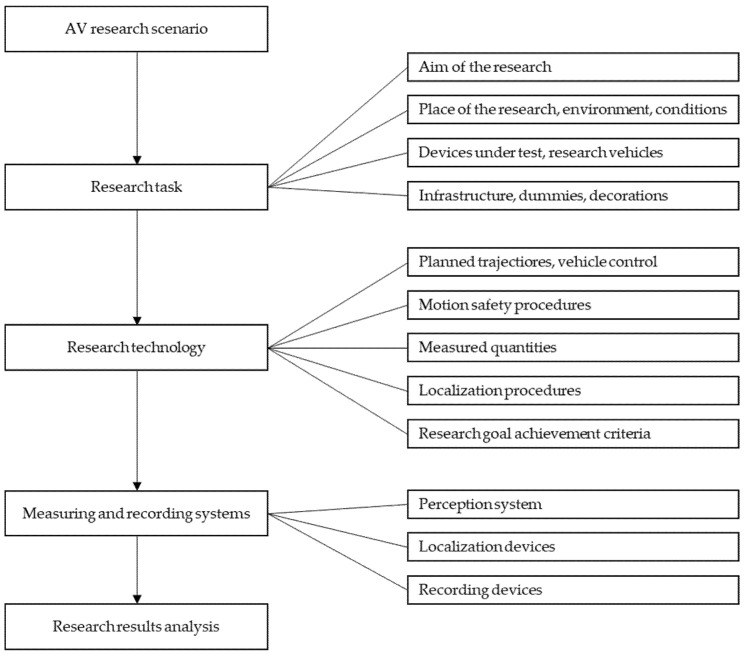
Hierarchical information diagram for AV research scenarios.

**Figure 2 sensors-22-06586-f002:**
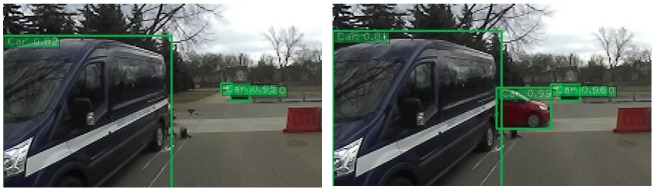
Part of the research: identification of the suddenly appearing obstacle at a road intersection by the AV’s perception system (Łukasiewicz Research Network—Automotive Industry Institute). Two frames from the perception system’s camera are shown; the numerical value in the figure indicates the probability of object identification by the AV system.

**Figure 3 sensors-22-06586-f003:**
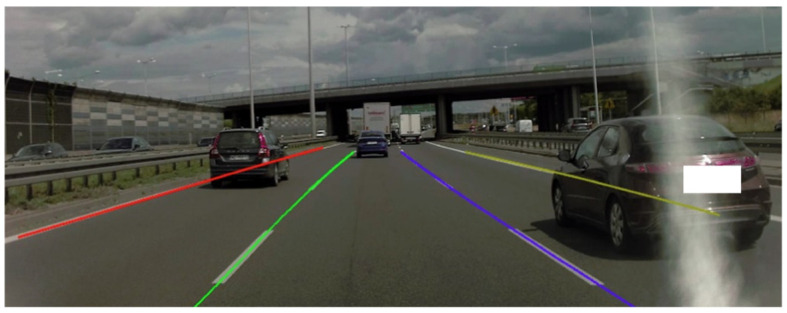
Part of the research: lane and road edge detection by the AV’s perception system (Łukasiewicz Research Network—Automotive Industry Institute).

**Figure 4 sensors-22-06586-f004:**
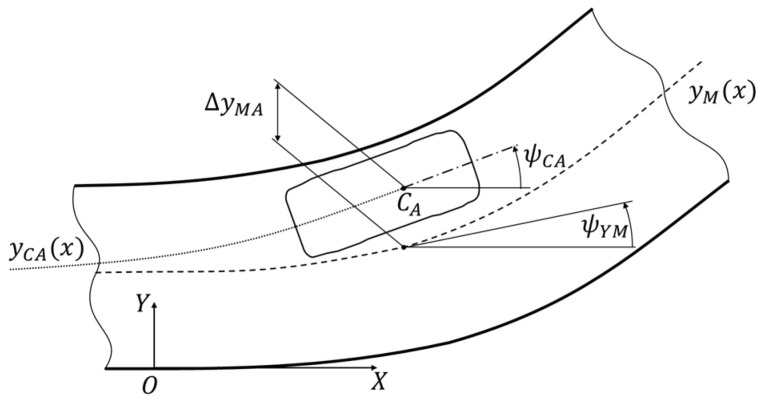
Example of the course of the planned driving path $y_{M}\left(x\right)$ and actual resultant trajectory $y_{CA}\left(x\right)$; the method for measuring lateral deviation ($\Delta y_{MA}$ ) and the longitudinal axis of the car’s angular deviation ($\Delta \psi_{MA}$ ) has been marked.

**Figure 5 sensors-22-06586-f005:**
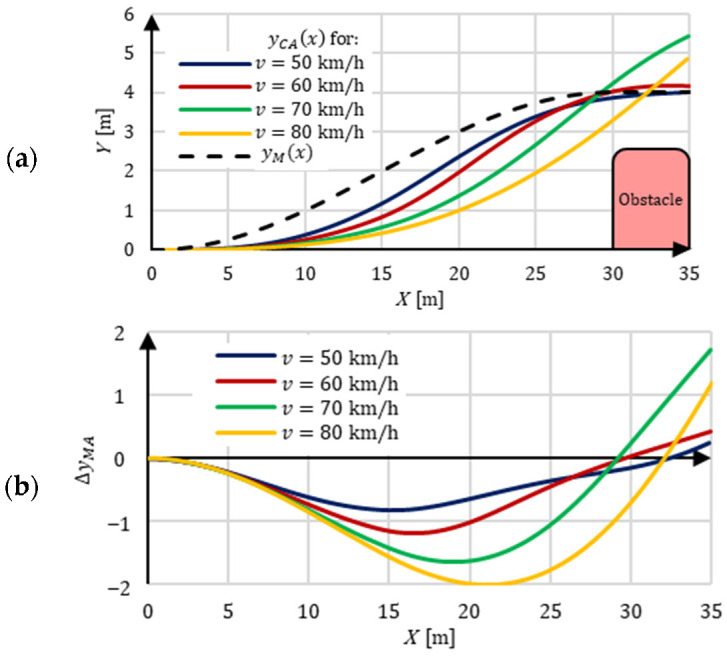
Planned driving path and resultant trajectory of the car for several driving velocities; v=50, 60, 70, 80 km/h; (**a**) planned driving path $y_{M}\left(x\right)$ and trajectory $y_{CA}\left(x\right)$; (**b**) lateral deviation $\Delta y_{MA}$ of the trajectory (movement of the center of mass) from the planned driving path; (**c**) longitudinal axis of the car’s angular deviation $\Delta \psi_{MA}$ from the tangent to the planned driving path.

**Figure 6 sensors-22-06586-f006:**
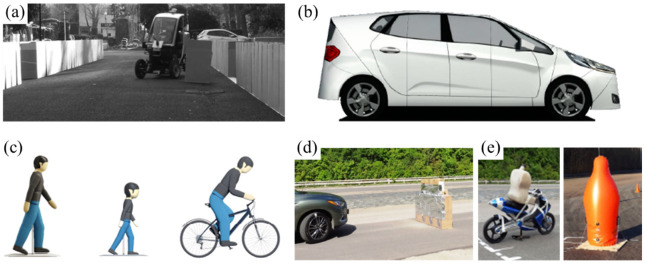
Obstacles examples used in experimental research scenarios: (**a**) pieces of cardboard to mark the road lane and the obstacle [34]; (**b**) soft car target on a moving platform [17,28]; (**c**) pedestrian, child, bicyclist dummy on a moving platform [43]; (**d**) soft wall covered with a metallized mirrored film [56]; (**e**) motorcyclist and human figure dummy [56].

**Table 1 sensors-22-06586-t001:** The main features of the test scenario for planning and maintaining the vehicle’s driving path [30,31,32].

**Task and aim of the research**	The research task in the described scenarios is automated driving along a detected road lane on a curvilinear track. The aim of the research is to keep the vehicle in the road lane. The lane is maintained by controlling the steered wheels or additionally by selecting the driving velocity.
**Research object**	The research used passenger cars equipped with systems enabling autonomous driving [30,31] or a go-kart which performs AV functions [32]. For this purpose, elements of the perception system, path planning, and the steering wheel control system were used. In this research, the path planning system determines the reference trajectory along the center line of the road lane.
**Research site/environment**	The location used for this research is specialized research tracks. KATRI [30], Aci-Sara Lainate and the Monza Eni Circuit [31] tracks were used. They create road infrastructure for safe and repeatable research in the form of closed loops. In [32], the perception system used track elements in the form of road signs to select driving velocity.
**Vehicle control system**	The control process was carried out by turning the vehicle wheels. The value of the steering angle results from the need to keep the vehicle on the road lane. The necessary correction takes into account the lateral and angular deviation (Figure 4 and Figure 5) between the planned path and the actual vehicle trajectory. The deviations are measured from the center line of the road lane [31] or the reference trajectory [30]. For example, the control process in [30] was carried out in a feedback loop, in which the lateral deviation of the vehicle from the planned path was taken into account. This deviation was the basis for calculating the steering angle of the vehicle’s wheels. An actuator with a stepper motor was used to make the turn.
**Additional information**	In [32], HIL technology was used, in which computer simulation uses the position of the road lines and the vehicle. The position of the vehicle was simulated according to the so-called bicycle model (CarSim—Matlab software). Images from the computer screen, observed through the camera, were the basis (in the feedback loop) for the evaluation of the position of the vehicle in relation to the computer simulation. The difference of these positions makes it possible to calculate the necessary correction in the form of the steering wheel. However, the camera is sensitive to shocks and even its minimal displacement makes it difficult to detect lines on the computer screen and calculate the correct deviation from the planned path.

**Table 2 sensors-22-06586-t002:** The main features of the test scenarios for the effectiveness of lane detection and keeping [27,33].

**Task and aim of the research**	The research task in the described scenarios is automated lane detection and keeping [27], as well as avoiding a slower moving vehicle [33]. The aim of the research has two aspects: effective detection of road lane boundary lines in conditions of traffic with other vehicles and determination of a non-collision (safe) trajectory while avoiding a slower car. In [27], a scenario was used in which the perception system was tested in terms of its effectiveness in detecting lines on the road. A more complex research task covers the scenario in [33]. The task additionally includes determining a non-collision path of movement so as to obtain smooth avoidance of a slower moving car.
**Research object**	Passenger cars and their measuring equipment were used during the research. Little information has been provided about the equipment of the vehicle in [27]. The scope of the planned research indicates that the vehicle perception system (camera and software) was involved.
**Research site/environment**	In both scenarios, research was planned in an urban area. In [33], a section of the road was simulated in a closed area (e.g., with dummies). In this area, a two-lane roadway was separated, which made it possible to avoid vehicles. In this area, driving velocities reached up to 25 km/h. In [27], there was a limited stretch of road traffic used in the scenario. The car’s perception system was used during analysis of the effectiveness of detecting the road lane lines.
**Vehicle control system**	During the research in scenario [27], attention was focused on the effective detection of lane boundaries in urban conditions. Vehicles in scenario [33] performed automated driving by turning the wheels and selecting driving velocity. The choice of velocity took into account the limitations of driving safety and smoothness.
**Additional information**	The IMU sensors used are inertial sensors, hence, when measuring the angle of deviation of the longitudinal axis of the vehicle, it is necessary to include the deviation value, which makes the result susceptible to the phenomenon of drift [33]. In scenario [27], a neural network was used to estimate the safe driving distance.

**Table 3 sensors-22-06586-t003:** The main features of the scenario including avoidance of suddenly appearing obstacles [34].

**Task and aim of the research**	The research task related to avoiding a suddenly appearing obstacle. This research is a reference for active safety technology aiming to prevent road accidents. A scenario was used that involved two aspects of research: obstacle avoidance and braking.
**Research object**	The research subject was a single-person AV that performed autonomous driving during obstacle avoidance. The functioning of perception, risk analysis, and obstacle avoidance path planning systems was investigated.
**Research site/environment**	The research was conducted in a closed area. Cardboard boxes were used to define the research area (roads and obstacles). A velocity limit of 30 km/h was introduced.
**Vehicle control system**	The vehicle was controlled by the braking and steering systems. The location of the edge of the road and the obstacle allowed for geometric location (in global coordinates) of the obstacle avoidance trajectory. This made it possible to calculate the necessary correction of the trajectory compared to the planned path in the vehicle control system.
**Additional information**	Cardboard boxes are often used to mark road boundaries and obstacles.

**Table 4 sensors-22-06586-t004:** Normative test scenarios for active safety and control systems in AVs.

Scope and Aim of the Research	Scenario Identification
Tests of the emergency braking system/collision warning system against an obstacle (vehicle, pedestrian, bicyclist). The aim of the test was to assess the effectiveness of a given AV system. Images from cameras and system interface signals were recorded. Measured values for the car: position, driving velocity, angular velocities, acceleration, steering wheel angular velocity, and the intensity of acceleration and braking.	Euro NCAP: AEB C2C (CCRs, CCRm, CCRb, CCFtap) [17], AEB VRU (CPFA, CPNA, CPNC [50,51], CPLA, CPTA, CPRA, CBNA, CBNAO, CBFA, CBLA) [43]
	ISO: 15623 [41], 19237 [52], 22078 [53], 22839 [54]
	IIHS: AEB [48], P-AEB [49]
	UNECE: 152 ] [38]
	RuNCAP: AEB [55,56,57,58]
Tests related to lane keeping, driver lane departure warnings, semi-automatic lane changes, and emergency lane keeping systems. The aim of the test was to assess the system’s effectiveness in different driving conditions (types of road lines and specificity of obstacles on the road). Images from cameras and system interface signals were recorded. Measured values for the car: position, driving velocity, angular velocities, and steering wheel angular velocity.	Euro NCAP: LSS (ELK, LKA, LDW) [28]
	SAE: J2808 [47], J3045 [16]
	ISO: 11270 [59], 17361 [40], 21202 [42]
	UNECE: 130 [37], 157 [39]
Tests of the system responsible for informing about speed limits and a system that adjusts driving velocity to road limits. The aim of the test was to assess the system’s effectiveness with different road types and driving velocities. Road infrastructure (road speed limit signs) and signaling of restrictions via the AV interface were recorded.	Euro NCAP: SAS (SLIF) [22]
	EU 2021/1958 [18]
Tests of the vehicle’s blind spot monitoring system and lane change support system. The aim of the test was to assess the system’s effectiveness in various road maneuvers. Images from cameras and system interface signals about lane departure hazards were recorded. Measured values for the car: driving velocity, acceleration, and steering wheel angle.	NHTSA: 812045 [44], 812317 [45]
	ISO: 17387 [60]
Tests of the driver warning system for excessive driving velocity on the curve of the road. The aim of the test was to assess the correctness of signaling related to excessive velocity on the curve of the road through the system interface. Arc radius value and overspeed signaling via the system interface were recorded. Measured values for the car: position and driving velocity.	ISO: 11067 [61]
Testing of the active cruise control system and the system that controls following the vehicle in front at low speeds (traffic jam assistant). The aim of the test was to assess the effectiveness of the system for different driving modes. Detected vehicles before the AV were recorded. Measured values for the car: AV motion parameters and distance to the obstacle.	ISO: 15622 [62], 22178 [63]
Testing of the assisted parking system, maneuvering aids system, and the system for predefined routes for low-speed operations. The aim of the test was to assess the effectiveness of parking area detection, detection of obstacles, scanning the space around the vehicles, path planning, and control. Images from cameras, information about obstacles, and signals from the scanning sensors were recorded. Measured values for the car: position and distance to the obstacle.	ISO: 16787 [64], 17386 [65], 22737 [66]

**Table 5 sensors-22-06586-t005:** Examples of artificial objects and their roles in the research scenario.

Aim of the Research	Artificial Objects	Role in the Research	Publication
Avoiding obstacles, braking in front of obstacles	Cardboard boxes	Marking a road lane	[34]
Control of the vehicle braking process	Dummy parts of the rear of a car body (stationary or movable)	Imitation of a car on the road	[72,93,94]
2D and 3D obstacle identification	Road cones	Objects to be identified by the perception system	[67]
Critical maneuvers to avoid collisions with suddenly appearing obstacles	Soft wall covered with a metallized mirrored film	Obstacle on the AV’s driving path (stationary, mobile)	[55]
Critical maneuvers to avoid front-end collisions with suddenly appearing obstacles	Pedestrian, child, and bicyclist dummies on a moving platform; soft car target	Moving objects on the path intersecting with the AV’s driving path	[51,56]
Defensive maneuvers before the collision; the lane change problem	Susceptible obstacle on a moving platform	Moving objects on the path intersecting with the AV’s driving path	[17,28]

**Table 6 sensors-22-06586-t006:** Properties of sensors in AV perception systems [20].

Parameter	Camera	Thermal Camera	Radar	Lidar
Resolution	Good	Good	Fair	Fair
Illumination	Poor	Good	Good	Good
Weather	Fair	Good	Fair	Good
Cost	Good	Fair	Poor	Poor

**Table 7 sensors-22-06586-t007:** Cameras used in perception and control systems during AV research.

Research Scope of the Scenario	Type, Manufacturer, Model, and Selected Sensor Parameters	Publication
Lane keeping by AVs	**Camera, Bosch** (first generation, CMOS).	[32]
3D obstacle detection, braking and avoiding the obstacle	**RGB camera** (H: 48 deg).	[72]
2D obstacle detection, localization and object tracking, lane/road detection	**Camera, Hitachi KP-F3** (R: 644 × 493 px.).	[70]
3D obstacle detection	**RGB camera** (H: 90 deg, V: 30 deg, R: 1920 × 640 px.).	[25]
2D and 3D obstacle detection, localization, tracking	**Stereo camera**.	[21,23,67,77]
2D and 3D obstacle detection, localization, tracking, research of emergency braking/collision warning systems (car, pedestrian, bicyclist)	**RGB camera.**	[20,26,50,68,69]
2D object detection, localization, tracking	**Thermal camera**.	[20]
Research of emergency braking/collision warning systems (car, pedestrian, bicyclist)	**Camera in the Subaru EyeSight system**.	[55]
Research of emergency braking/collision warning systems (car, pedestrian, bicyclist)	**Camera in the Infinity FEB system**.	[56]
Research of emergency braking/collision warning systems (car, pedestrian, bicyclist), research of the lane keeping/lane departure warning/semi-automatic lane change/emergency lane keeping systems, research of blind spot monitoring/lane change assist systems	**RGB camera**.	[17,28,43,44,51]
Lane keeping by the AV, 3D object detection	**Stereo camera, Stereolabs ZED** (R: 672 × 376 px., F: 100 Hz, Range: 25 m).	[31,79,96]
Lane/road detection, 3D object detection	**Camera, Sony PC-350**.	[27]
3D object detection	**Camera, Sekonix SF3321** (f: 30 Hz).	[24]
2D and 3D object detection, localization, tracking	**Camera, Logitech HD Pro C920** (R: 1920 × 1080 px., f: 30 Hz); **Camera, Giroptic HD** (H: 360 deg, R: 2048 × 1024 px., f: 30 Hz); **Thermal camera, FLIR A320** (R: 380 × 240 px., f: 9 Hz); **Thermal camera, FLIR A65** (R: 640 × 512 px., f: 30 Hz); **Thermal camera, Tonbo Imaging Inc. HawkVision** (R: 640 × 480 px., f: 25 Hz); **Camera, New Imaging Technology NSC1003** (CMOS, R: 1280 × 1024 px., f: 25 Hz); **Camera, Point Grey Two Flea/FL3-GE-28S4C-C** (R: 1928 × 1448 px., f: 15 Hz); **Camera, Carnegie Robotics Multi Sense** S21 (f: 30 Hz).	[95]
2D and 3D object detection, localization, tracking	**Camera** (H: 85 deg, R: 3840 × 2160 px., f: 30 Hz).	[75]
2D and 3D object detection, localization, tracking	**Stereo camera, Point Grey Bumblebee XB3 (BBX3-13S2C-38)** (R: 1280 × 960 px., f: 16 Hz, H: 66 deg); **Camera, Point Grey Grasshopper2 (GS2-FW-14S5C-C)** (R: 1024 × 1024 px., f: 11 Hz, H: 180 deg).	[74]
2D and 3D object detection	**Logitech StreamCam** (R: 1280 × 720 px., f: 60 Hz, H: 78 deg)	[97]
3D obstacle detection	**Leopard Imaging AR023ZWDR**	[89]

**Table 8 sensors-22-06586-t008:** Radars used in perception and control systems during AV research.

Research Scope of the Scenario	Type, Manufacturer, Model, and Selected Sensor Parameters	Publication
Braking and avoiding the obstacle	**Short-range radar** (H: 30 deg, Range: 30 m, f: 76.5 GHz); **Long-range radar** (H: 20 deg, Range: 150 m, f: 76.5 GHz).	[71]
Braking and avoiding the obstacle, 3D object detection	**Radar, Fujitsu Ten** (Range: 120 m, H: 16 deg); **Radar, Mitsubishi** (Range: 150 m, H: 12–16 deg); **Radar, Denso** (Range: 150 m, H: 20 deg); **Radar, Nec** (Range: 120 m, H: 16 deg); **Radar, Hitachi** (Range: 120 m, H: 16 deg); **Radar, A.D.C.** (Range: 150 m, H: 10 deg); **Radar, Bosch** (Range: 150 m, H: 8 deg); **Radar, Autocruise** (Range: 150 m, H: 12 deg); **Radar, Delphi** (Range: 150 m, H:16 deg); **Radar, Eaton** (Range: 150 m, H:12 deg); **Radar, Visteon** (Range: 150 m, H: 12 deg).	[72]
Research into emergency braking/collision warning systems (car, pedestrian, bicyclist)	**Radar, Continental ARS510** (H: 9 deg, Range: 120 m); **Radar, Continental SRR510** (H: 90 deg, Range: 30 m).	[17,43,50,51,55,57]
Research into emergency braking/collision warning systems (car, pedestrian, bicyclist)	**Radar in the Infinity FEB system**.	[56]
3D object detection	**Radar, Delphi ESR 64**.	[95]

**Table 9 sensors-22-06586-t009:** Lidars used in perception and control systems during AV research.

Research Scope of the Scenario	Type, Manufacturer, Model, and Selected Sensor Parameters	Publication
Planning the driving path of an AV, 3D object detection	**Lidar, Velodyne HDL-64E** (H: 360 deg, Layers: 64, Range: 120 m).	[23,24,25,33,79]
Planning the driving path of an AV, braking and avoiding the obstacle	**Lidar, Sick.**	[34]
Braking and avoiding the obstacle, 3D object detection	**Lidar, Mitsubishi** (H: 12 deg, V: 4 deg, Range: 130 m); **Lidar, Denso** (H: 16 deg, V: 4.4 deg, Range: 120 m); **Lidar, Denso** (gen.II, H: 40 deg, V: 4.4 deg, Range: 120 m); **Lidar, Nec** (H: 20 deg, V: 3 deg, Range: 100 m); **Lidar, Omron** (H: 10.5 deg, V: 3.3 deg, Range: 150 m); **Lidar, Omron** (gen.II, H: 20–30 deg, V: 6.5 deg, Range: 150 m); **Lidar, Kansei** (H: 12 deg, V: 3.5 deg, Range: 120 m); **Lidar, A.D.C.** (H: 17 deg, Range: 150 m).	[CYT42] [72]
3D object detection	**Lidar, Velodyne HDL32E** (H: 360 deg, Layers: 32, Range: 100 m, f: 10 Hz).	[67,78,95]
3D object detection	**Lidar**, (H: 90 deg, Layers: 64).	[68]
3D object detection	**Lidar, Ibeo Lux 8** (H: 110 deg, Layers: 8, Range: 50 m).	[76]
Research into blind spot monitoring/lane change assist systems for Avs	**Lidar**.	[44]
3D object detection	**Lidar, Velodyne VLP-16** (H: 360 deg, Layers: 16, Range: 100 m).	[79,97]
3D object detection	**Lidar, Hesai Pandar 40p** (H: 360 deg, V: 40 deg, Layers: 40, Range: 200 m, f: 10 Hz).	[75]
3D object detection	**Lidar SICK LMS-151 2D** (H: 270 deg, f: 50 Hz, Range: 50 m). **Lidar SICK LD-MRS 3D** (H: 85 deg, V: 3.2 deg, Layers: 4, f: 12.5 Hz, Range: 50 m).	[74]
Planning the driving path of an AV, 3D object detection	**Lidar Sick LD MRS** (H: 94 deg, Layers: 8, Range: 200 m).	[19]
3D obstacle detection	**Lidar Velodyne VLS-128** (H: 360 deg, Layers: 128, Range: 245 m).	[89]

**Table 10 sensors-22-06586-t010:** Other sensors used in perception and control systems during AV research.

Research Scope of the Scenario	Type, Manufacturer, Model, and Selected Sensor Parameters	Publication
Planning the driving path of an AV, 3D object detection	**GPS**.	[19,33]
Planning the driving path of an AV, braking and avoiding the obstacle	**IMU**.	[33,34]
Lane keeping by the AV, 2D and 3D object detection, localization, tracking	**GNSS RTK SwiftNavigation**.	[31,75]
2D and 3D object detection, localization, tracking	**IMU Vectornav VN-100**. **Single RTK GPS Trimble AG GPS361**. **Differential RTK GPS Trimble Dual Antennas A BD982**.	[95]
2D and 3D object detection, localization, tracking	**OXTS RT-Range.**	[75]
2D and 3D object detection, localization, tracking	**GPS/INS NovAtel SPAN-CPT Align inertial and GPS navigation system**. **GPS/GLONASS dual antenna**.	[74]
3D object detection	**IMU Xsens Mti-G**.	[19]
2D and 3D object detection	**RTK GNSS GPS MRP-2000.**	[97]
3D obstacle detection	**GPS/IMU with RTK Novatel PwrPak7 dual antenna.**	[89]

## Data Availability

Data sharing not applicable.

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
