# Peer review of "Research Scenarios of Autonomous Vehicles, the Sensors and Measurement Systems Used in Experiments"

_sensors, 2022, doi:10.3390/s22176586_

Round 1
Reviewer 1 Report
The paper provides a comprehensive review on different aspects of research on AVs. Research scenarios are categorized considering different aims and objectives, features, hardware, etc.
The paper is timely, well organized and well written and sill be interesting for the readers and researchers in the field. It also covers all important recent and relevant research articles. I do not have any major comments on the paper, just one suggestion and a minor comment:
1- A general chart to overview different aspects of a research on AV, which are covered in the paper, is very handy and informative. It can be a multi-layer hierarchical chart with, for example, "Technical aspects" as a parent and "Research Vehicles", "Research sites", "Infrastructure", and titles of other subsections in section 4 as children.
2- In page 3, line 115, Do you mean "EU"?
Author Response
The authors would like to thanks for the thorough analysis of the content of our article. The comments from the Reviewer has been used to prepare revised version of the article. Authors' answers for the comments are as follows:
1 - On the basis of the literature analysis, a hierarchical scheme of information necessary for creation of AV research scenarios has been built. The layout of this scheme is shown in Figure 1 in the article (page 4 line 204). The article presents individual elements of this set.
2- Thank you for paying attention to this abbreviation. Naturally it should be "EU" and this error has been corrected in the article.
Reviewer 2 Report
In this paper, the review is carried out for finding out scenarios used in the existing literature within the context of automated and autonomous vehicles. The idea of conducting research in this direction is interesting, however, there are some major issues that need to be resolved before accepting this paper for publication.
- - The authors have not provided the details about how the review process was conducted. Is there any systematic process that has been followed for performing the review, such as the usage of databases for extracting materials? Applying some quality assessments and defining research questions, etc.?
- Are there any selection criteria used for selecting and retrieving information from existing literature?
- - The purpose or need for conducting the review is not clear. Who is the target audience? Who can get benefits from this review?
- - What are existing gaps in the literature/selected research work and what are the future research directions that someone can consider? The authors only discussed the information present in the literature. But have not criticized them properly, i.e., what is not considered.
- - The authors have not discussed the limitation(s) and threats to the validity of this review. This should be presented in a separate section.
Author Response
The authors would like to thanks for the thorough analysis of the content of our article. The problems noticed by the Reviewer were used to prepare the revised version of the article. We addressed all of the Reviewer's comments. The answer to each of them is provided below.
- The authors have not provided the details about how the review process was conducted. Is there any systematic process that has been followed for performing the review, such as the usage of databases for extracting materials? Applying some quality assessments and defining research questions, etc.?
- Are there any selection criteria used for selecting and retrieving information from existing literature?
Collecting research material for the article was based on the use of many scientific publications databases. The analysis was carried out with the use of Springer, Science Direct, MDPI, Taylor & Francis, IEEE Xplore, ResearchGate, SAE International databases. The online resources of the Euro NCAP; NHTSA; ISO; UNECE; IIHS organizations were analyzed separately. The search process has been conducted continuously for several years. It is based on a constantly updated set of keywords. Currently it is: 2D / 3D object detection (car, pedestrian, bicyclist, sign); Active safety; Advanced Driver Assistance Systems (ADAS); Autonomous emergency braking (AEB); Artificial intelligence; Autonomous vehicles accidents; Autonomous vehicles / cars (AV); AV sensors; Camera detection; Collision avoidance; Collision risk; Computer vision; Control; Decision-making; Deep learning; Distance estimation; Emergency steering / braking; Experimental research; Fuzzy logic; Hardware in the loop (HIL); Image; Lane change maneuver; Lane departure warning system (LDWS); Lidar detection; Line detection; Lane keeping assist (LKA); Model predictive control (MPS); Neural network; Object detection metrics (precision, recall, IOU); Obstacle detection; Path planning, tracking, following; Perception system; Point cloud; Radar detection; Rear-end collision avoidance; Research / test scenarios; Road / traffic safety; Safety distance; Fusion sensor; Software in the loop (SIL); Stereo vision / cameras; Suddenly appearing obstacle; Vehicle dynamics; Vehicle stability.
The publications selected in this way were assessed in terms of meeting the criteria:
- the relationship of keywords with the topic of this article;
- the content of the results of AV experimental research;
- the presence of information on the research scenario.
The auxiliary criteria for the evaluation of the publication were the journal's IF number, citations, number of recommendations and number of reads, and the year of publication.
During the literature review, the authors looked for answers to many questions. The most frequently questions were:
- What scenarios were considered in the AV research?
- Was a critical situation considered during the research? Which one?
- Did the research scenario take into account the normative documents for the AV research procedures?
- Where were the AV resarch (road traffic, test track / proving ground) carried out?
- Was the entire vehicle or component tested?
- What components of the AV system were tested?
- What control system was used during the research?
- What type of vehicle was tested (passenger car, truck, other)?
- What research task was taken into account in the analyzed scenario?
- What sensors and perception system were used during the research and what are their basic technical parameters?
- What locatization devices were used?
- What other measuring and recording equipment has been used in the research?
Answers to the above research questions were collected in a structured form of an own database. The material structured in this way was the starting point for the development of the submitted article. A synthetic description of the process of collecting the article material supplemented the content of the work in Chapter 1 (page 5 lines 206-213).
- The purpose or need for conducting the review is not clear. Who is the target audience? Who can get benefits from this review?
After a short description of the main problems of AV participation in road traffic and their research, the scenarios of experimental AV research as well as the measuring and recording equipment and systems used were presented.
The aim of the article is to present the main information about the AV research scenarios and the research tasks undertaken in this area in an orderly manner. The collected material aims to systematize information that will facilitate planning and conducting experimental AV research. The developed material will allow to shorten the time of their preparation, avoid repeating the tests already performed, reduce the number of wrong decisions at this stage (selection of the research task, measuring and recording equipment, location devices, etc.). Systematizing the available information on normative and nonnormative research scenarios will facilitate their analysis and the assessment of their applicability when planning AV studies.
The material gathered in the article will help answer the following questions:
- research tasks that currently dominate;
- elements to be considered in the AV research scenario;
- which normative documents are useful in organizing the planned research;
- what measuring and recording equipment is necessary?
The article summarizes the currently used normative scenarios of experimental AV research. The scenarios used by various research organizations were characterized, in which the plan and course are adapted to the current needs of researchers (nonnormative scenarios). Attention was also focused on the technical aspects of the research scenarios and the measuring systems and sensors used in the research.
The target group of the conducted literature review are people who are interested in the issue of AV research and who organize experimental vehicles research for exploratory purposes and to improve their design. There are no results in the literature that provide structured information on the organization and scenarios of the AV research. The article indicates the currently dominant research tasks and methods of AV testing. An important area of ​​systematized information are also lists of standards describing standardized procedures for ADAS systems testing. In addition, an important part of the article is the list of frequently used sensors conected to the currently available AV tests. This makes it possible to compare the results of own research with the use of a given sensor model to the results of other works in this field.
The article contains a supplement to the aim of the work (page 4, lines 174-186) and information on the benefits of this work (chapter 6, lines 768-773).
- What are existing gaps in the literature/selected research work and what are the future research directions that someone can consider? The authors only discussed the information present in the literature. But have not criticized them properly, i.e., what is not considered.
There is a lack of publications focused the reader's attention on the preparation and organization of AV research. In publications that contain the results of experimental research, the authors focus on the results of these studies, and there is very little information about its preparation and scenario. This additionally justifies the need to gather and systematize knowledge in this area.
Developing AV research is important, necessary, and useful. Road traffic tests, including obstacle avoidance and the steering system response in critical situations, are of great importance. Thus, the scenarios should increasingly take into account real road traffic, concidering a variety of upsets. Most of them concern incidents at a road junction. There are critical situations in keeping a curvilinear lane while driving at high speed. There are no scenarios where the complex AV control process is tested, taking into account turning, braking and acceleration in common.
There are few publications with the results of experimental AV research, because we are at the initial stage of their development. Such research is difficult, risky and very costly. Hence, they are largely conducted in automotive concerns. They have the necessary budget, technology and research + development facilities. This area is dominated by normative research aimed at the approval and certification of new proprietary solutions, especially for commercial purposes. Few of such studies are published. Information on research scenarios is particularly limited. With a small number of publications, it is difficult to indicate the advantages and disadvantages of the analyzed scenarios. It is difficult to criticize unit solutions (in terms of research scenarios), because they are largely unique.
Therefore, the main goal at this stage is to collect and systematize knowledge so that it is easily available when planning subsequent experimental research, especially of cognitive and developmental importance.
-The authors have not discussed the limitation(s) and threats to the validity of this review. This should be presented in a separate section.
The limitations in the conducted analysis result from several reasons:
- there is a lack of publications reporting experimental research of AV;
- there is little information that can be obtained about the experimental research sce-narios in the analyzed publications;
- the lack of many legal regulations in the field of these studies means that a large part of the scenarios should be treated as individual ideas and solutions.
- many researchers treat the matters of the research scenario as a whole of experience, practice and procedures, or otherwise as details of their research workshop, which are their own know-how and do not feel the need to disseminate them.
The continuous development of driver assistance systems (ADAS) and AV influences the dynamic development of this automotive area. This will result in new publications that will enrich our knowledge in these areas. The technology of perception systems is expected to undergo intensive development. These systems will facilitate the process of automating vehicles and reduce the risk of a road accident and its consequences in the legal aspect.
The following points can be taken as the benefits of this publication:
- facilitates the organization of AV research, indicating the currently dominant study tasks and the sets of measuring and recording equipment used;
- allows to systematize knowledge about normative research and indicates beneficial ways of their implementation;
- facilitates the search for new, previously undeveloped areas in AV research, which will affect the dynamic development of autonomy systems in vehicles;
- provides information on the widely used sensors of AV systems from different manufacturers and the available research results with their participation.
The undertaken topic refers to the difficulties and experiences of the authors at the stage of organizing the preparation of own experimental tests of AV.
Synthetic information from the answer on this question supplemented the article content (page 22 lines 753-759).
Round 2
Reviewer 2 Report
The article has been updated with most of the reviewer's comments. However, Figure 1 is not readable, the quality of Figure 1 should be improved.